# Association of NK Cells with the Severity of Fibrosis in Patients with Chronic Hepatitis C

**DOI:** 10.3390/diagnostics13132187

**Published:** 2023-06-27

**Authors:** Anna Kleczka, Bogdan Mazur, Krzysztof Tomaszek, Andrzej Gabriel, Radosław Dzik, Agata Kabała-Dzik

**Affiliations:** 1Department of Pathology, Faculty of Pharmaceutical Sciences in Sosnowiec, Medical University of Silesia in Katowice, Ostrogórska 30, 41-200 Sosnowiec, Poland; adzik@sum.edu.pl; 2Department of Microbiology and Immunology, School of Medicine with the Division of Dentistry in Zabrze, Medical University of Silesia in Katowice, 40-808 Zabrze, Poland; bmazur@sum.edu.pl; 3Department of Pathomorphology, School of Medicine with the Division of Dentistry in Zabrze, Medical University of Silesia in Katowice, 40-800 Zabrze, Poland; krztomaszek@gmail.com (K.T.);; 4Faculty of Biomedical Engineering, Department of Biosensors and Processing of Biomedical Signals, Silesian University of Technology, Roosevelta 40, 41-800 Zabrze, Poland; radoslaw.dzik@gmail.com

**Keywords:** NK cells, hepatic stellate cells, fibrosis, chronic hepatitis C, flow cytometry

## Abstract

Some NK cell subpopulations may be involved in the modulation of fibrogenesis in the liver. The aim of the study was to evaluate the relationship between the number and phenotype of NK cell subsets in peripheral blood (PB) and total NK cell percentage, population density and the degree of liver fibrosis of patients infected with hepatitis C virus (HCV+). The study group consisted of 56 HCV+ patients, divided into two subgroups: patients with mild or moderate fibrosis and patients with advanced liver fibrosis or cirrhosis (F ≥ 3 in METAVIR classification). The preparations were stained with H-E and AZAN staining. NK cells were targeted with anti-CD56 antibody and identified automatically in situ using the DakoVision system. Assessment of different NK cell subsets in PB was performed with the flow cytometry technique. In the PB of HCV+ patients with advanced liver fibrosis, there was a lower proportion of CD62L+; CD62L+/CD94++; CD27+; CD127+/CD27+ and CXCR3+/CD27+ NK subsets, as compared to patients with mild/moderate liver fibrosis. The results also showed no association between total PB NK cell level and total intrahepatic NK cell population density between patients with mild/moderate fibrosis and with advanced liver fibrosis. However, positive correlations between the PB levels of CD94+ and CD62L+ NK cell subsets and the intrahepatic total NK cell percentage and population density in the liver, irrespectively to the extent of fibrosis, were observed. Additionally, positive correlation was found between the PB CXCR3+/CD94+ NK cell percentages and intrahepatic NK cell percentages in patients with advanced hepatic fibrosis. Lower blood availability of specific NK subsets in patients with chronic type C hepatitis might be a cause of progression of liver fibrosis via insufficient control over hepatic stellate cells.

## 1. Introduction

The World Health Organization (WHO) considers hepatitis C to be one of the greatest global epidemiological threats [1]. The source of infection is the blood of an infected person transferred to the bloodstream of a healthy person. In addition to the transfusion of infected blood (about 7.4% of the infection rate), transmission also occurs as a result of tissue disruption, e.g., during surgical and dental procedures (about 9.5%) and beauty treatments such as tattooing and piercing (about 8.4%). Most often, infections occur among intravenous drug users (approximately 58.6% of infections). Infection can also be transmitted through the placenta from an infected mother to the fetus, through sexual intercourse and through the sharing of personal hygiene products [2,3].

After the infection, the course of the disease is usually asymptomatic. The most common symptoms of infection are non-specific symptoms (e.g., muscle and joint pain, fatigue), which, combined with unawareness of infection, mean that the detection of the disease is usually accidental and late. Symptomatic acute hepatitis C associated with jaundice occurs in approximately 20–30% of patients, approximately 3–12 weeks after infection. In about 15–25% of patients, the immune mechanisms are able to fight off the virus infection on their own. This requires a lively, polyclonal humoral response against the rapidly mutating and antigen-changing virus. In most patients, viral infection becomes chronic, which is associated with serious consequences-increasing liver fibrosis leading to cirrhosis, and an increased risk of developing hepatocellular carcinoma [4].

Liver fibrosis is the body’s response to long-term exposure to a damaging factor. The stellate cells begin to produce an excessive amount of connective tissue, which results in progressive fibrosis of the extracellular matrix, followed by loss of hepatocyte communication with blood vessels and bile ducts. In cirrhosis, the histological structure of the liver is completely obliterated. 

The stage of fibrosis (staging) is scored. The most frequently used scale is the METAVIR (Meta-analysis of Histological Data in Viral Hepatitis) scale, which distinguishes five degrees of liver fibrosis. In degree F0, the presence of collagen fibers in the liver is not observed. Grade F1 is found with non-septal portal fibrosis and graded as mild fibrosis. F2 is characterized by portal fibrosis with few septa and F3 is described as multiple septa, bridging fibrosis without regenerative nodules. The highest METAVIR score (F4) indicates probable or definite cirrhosis [5,6,7].

A similar scale is assumed by Scheuer’s fibrosis classification system. It is distinguished by stage 0—no fibrosis; stage 1—with fibrosis in the portal spaces without septa; stage 2—described as moderate fibrosis, in which the forming fibrotic septa penetrate into the lobule but do not reach the central veins and do not form bridges between the portal spaces. Advanced fibrosis is found in stage 3, where there are already septa between adjacent portal spaces and portal spaces and central veins. Grade 4 of the Scheuer classification means cirrhosis [8].

The main role in the pathomechanism of fibrosis is played by hepatic stellate cells (HSC), which, as a result of activation, turn from cells storing vitamin A in Disse spaces into myofibroblasts. Activation consists of two main stages: initiation and maintenance, and expiration. As a result of initiation caused by exposure to elements of disintegrating hepatocytes, HSCs go into an activated state, and as a result their behavior changes—cells begin to divide, show chemotaxis and contraction, produce collagen fibers, break down the extracellular matrix, as a result of which the amount of collagen I and III increases and retinoids are lost and chemokines secreted (including CCL2, CCL3, CCl5, CXCL1, CXCL9 and CXCL10). As a result of activation, the number of CXCR3, CCR5 and CCR7 receptors on the surface of HSCs is increased, among which CCR5 activation especially affects the migration and proliferation of HSCs. The opposite function is demonstrated by the action of the chemokine CXCL9 through its receptor CXCR3 [9,10,11,12]. 

Activated HSCs secrete pro-inflammatory chemokines that interact directly with cells of the immune system, e.g., with natural killer cells. NK cells, depending on the expression of different surface markers, can engage in different cellular pathways and perform different, sometimes opposing, functions. NK cells have obvious cytotoxic activity, eliminating tumor cells and virus-infected cells (CD56dimCD16+ subpopulation) [13]. Some of them have prolonged persistence and antitumor activity in vivo (CD62L+ NKT cells) [14]. The density of the CD 94 antigen on the surface of CD56bright and CD56dim cells can alter the activity of natural killer cells by differential expression of granzyme B, perforin and IFN-γ secretion [15]. The CXCR3 receptor, in turn, is involved in the migration of NK cells and their suppressive effect on T lymphocytes. Research is underway on the role of this molecule in the effectiveness of antiviral vaccinations [16,17].

NK cells exhibit anti-fibrotic activity by inhibiting or directly destroying stellate cells. NK cells induce apoptosis by producing IFN-ɣ. Aroused stellate cells also reduce the expression of HLA-I surface proteins, which increases their susceptibility to NK cells. The relationship between NK cells and stellate cells is regulated by T-reg cells, which have an inhibitory effect on NK cells. Among the NK phenotypes, a specific subtype of CD56+CXCR3+ cells has been demonstrated, and these cells exhibit increased activity directed against stellate cells, thus inhibiting the fibrosis process. Factors that change the activity of NK cells also include vitamin A derivatives, stored by HSCs and lost as a result of their activation through autophagy. Retinol released from HSCs has a suppressive effect on NK cells, and retinoic acid—formed as a result of the expression of collagen and TGFβ proteins—stimulates the cytotoxic effect of NK cells on HSCs. A greater tendency to fibrosis was also observed in the case of hepatocyte steatosis [18,19,20].

Due to the expression of tetraspanin CD81 on the surface of natural killer cells, the E2 envelope protein of HCV, which is structurally similar to it, exhibits antiviral blocking activity and disrupts migration activity, which may facilitate the maintenance of chronic inflammation [21,22]. In addition, HCV core proteins impair NK function by p53-dependent upregulation of TAP1 and surface MHC class I expression. It has also been shown that some HCV core peptides stabilize HLA-E surface expression, which reduces NK cytotoxicity by acting on the CD94/NKG2A receptor. NK cells obtained from HCV-infected patients have an increased expression of the CD94 receptor, which in combination with increased secretion of IL-10 and TGF-beta results in the reduced ability of NK cells to activate dendritic cells [23,24,25].

The aim of the study is to evaluate differences in the percentage of natural killer cells and their subtypes in peripheral blood in patients with CHC between the groups with mild or moderate fibrosis and with advanced hepatic fibrosis. (≥F3). An advanced level of liver fibrosis was defined as lesions above ≥F3 in the METAVIR classification. In addition, the correlation of NK phenotypes in peripheral blood with the percentage and population index of intrahepatic NK cells depending on the stage of fibrosis will be assessed.

## 2. Materials and Methods

### 2.1. Patient Characteristics

The study included patients with CHC (Chronic Hepatitis C) who underwent a diagnostic biopsy to qualify for treatment with interferon and ribavirin preparations at the “ID Clinic” in Mysłowice, where, after obtaining histopathological evaluation of the liver biopsy, they received treatment. The qualification of patients for the experiments lasted from 2014 to 2017. Following the guidelines of The European Association for the Study of the Liver (EASL), it was assumed that the identification of advanced fibrosis or cirrhosis, serum biomarkers/scores and/or transient elastography is less accurate, and it is important to confirm these advanced stages using liver biopsies. Patients with confirmed fibrosis and a positive RT-PCR result for the presence of HCV RNA were referred for dual-therapy interferon and ribavirin [26]. Nowadays, the WHO (World Health Organization) recommends the treatment of hepatitis C with the use of DAA (direct-acting antivirals) [27]. The study included a group of 56 patients who were confirmed to be HCV infected using a RT-PCR test and who were qualified for interferon treatment in Silesian Voivodship (Poland). All patients were over 18 years of age and fully consciously agreed to participate in the medical experiment. Before the study, each patient was informed about taking part in the experiment and presented with information on the risks, subsequently giving consent to participate in the study. The research was approved by the bioethics committee in Resolution No. KNW/0022/KB1/130/13 of 17 December 2013.

In 15 of the subjects, the degree of liver fibrosis was defined as advanced or cirrhosis (grade F3 or F4 on the METAVIR scale). There were 8 men and 7 women aged 58.87 ± 9.31 years. In the remaining 41 patients (24 women and 17 men with an average age of 44.39 ± 12.65), minimal or low-stage liver fibrosis was found, classified as F0, F1 or F2 in the METAVIR scale. The analysis of gender distribution and group membership confirmed the independence of groups from this feature. There was no statistically significant relationship between sex and age. However, the statistical analysis revealed that the patients with advanced liver fibrosis (≥F3) are significantly older than the patients with mild or moderate fibrosis.

Underage patients, alcohol abusers (the exclusion criterion was consumption of >20 mg of pure alcohol/day in women, >50 mg of pure alcohol/day in men) and those taking hepatotoxic drugs, including steroids, were excluded from the study. In addition, those with pregnancy, diabetes, atherosclerosis or renal failure, as well as those with a diagnosed autoimmune disease, hyperthyroidism or cancer, were disqualified from the study. Patients with co-infections of HIV and HBV were also excluded from participation in the experiment. The criteria for classifying/excluding patients from the experiment are presented in Figure 1.

### 2.2. Tissue Material

#### 2.2.1. Histological Preparation

Liver biopsy was performed to qualify patients for antiviral treatment using the Hepafix kit (B.Braun, Melsungen AG, Germany). The tissue material was fixed in 5% buffered formalin, performed in the Milestone HISTOS5 tissue processor without the use of microwaves. Then, sections of 5 μm were cut from the resulting paraffin curls. A mechanical rotating microtome was used. Slides were routinely stained with hematoxylin and eosin, and with AZAN staining to visualize collagen fibers. For further immunohistochemical determinations, six serial preparations were cut from each block (some preparations were used in another study). 

Immunohistochemical staining was performed manually by an experienced laboratory diagnostician. Immunohistochemical slides were dewaxed, hydrated, and the antigens were discovered via boiling in a solution of 0.01 M sodium citrate for 20 min. The CD56 monoclonal antibody (DAKO, Glostrup, Denmark, M7304) was used as the primary antibody. Antigen–antibody reactions were visualized with the DakoVision system (DAKO, Glostrup, Denmark) using 3,3-diaminobenzidine as a chromogen. The sections were then stained with Mayer’s hematoxylin. The tonsil was used as a positive control, repeating the control with each staining batch. 

#### 2.2.2. Histopathological Evaluation 

The H&E and AZAN preparations were evaluated in the Olympus BX43 light microscope to determine the inflammatory activity and the severity of fibrosis according to the Scheuer and Metavir scales [28]. The assessment was carried out by a specialist pathologist with experience in liver pathology and was performed without access to information about the results of the cytofluorometric analysis. The figure shows the classification of fibrosis depending on the severity of collagen fibroplasia determined via AZAN staining. 

#### 2.2.3. Digitization and Digital Analysis 

The obtained preparations were scanned using the Histech MIDI 3D scanner. Digitized preparations were calibrated relative to the pixel size; this allowed for the determination of the surface area of a selection with an irregular surface. The digital image also allowed for quantitative and qualitative analysis, allowing for accurate, numerical determination of the number of areas with a given color and color intensity. It was possible to exclude areas of a certain shape and size from the analysis [29,30]. 

Numerical analysis was carried out on the scans using the QuantCenter analytical software by 3D Histech (version 2.3. by 3DHistech, Budapest, Hungary), using the HistoQuant and NuclearQuant modules. Fields of Interest (AOIs—Automated Optical Inspections) were determined using a WACOM Cintiq 12WX graphics tablet that allows for accurate contouring of fields using the manual selection mode in CaseViewer 3dHistech software (version 2.4 by 3DHistech, Budapest, Hungary). Each portal space, outlined along the contours of the endplate, was considered as an AOI, including the foci of bite necrosis and crossing the endplate via inflammatory infiltration. The edge of the section where a diffusion reaction was observed that could disturb the operation of the counting module was also excluded from the area under analysis. All portal spaces present in the preparation were analyzed. 

Digital image analysis was performed using two QuantCenter 3DHistech modules. The NuclearQuant module for counting nuclear reactions has shown its usefulness in counting positive reactions involving marginally truncated NK cells, resulting in a ‘pseudonuclear’ color reaction, and in counting the total number of lymphocytes. To illustrate the reaction, the following colors were used: blue for lymphocytes and yellow and red for cells with a positive CD56 reaction. Single bile ducts present within the portal space had little effect on the total number of lymphocytes counted within the portal space. 

The sum of the cells counted by the NuclearQuant module was taken as the total number of lymphocytes for the percentage analysis. For the analysis of the population density of CD56+ cells, the number of positive cells in relation to the selection area on which the analysis was carried out was adopted, the result value being presented as the number of cells per 10,000 µm^2^ of the surface.

In addition, the lymphocyte population density factor was taken into account for the analysis, which is the ratio of the total number of lymphocytes obtained from the NuclearQuant module to the AOI area, expressed as the number of lymphocytes per 10,000 µm^2^ of surface.

### 2.3. Cytofluorometric Analysis 

Whole peripheral blood collected in an EDTA solution in the amount of 5 mL was transported on the day of collection to the Immunology Department of the SP PSK1 Laboratory in Zabrze for cytofluorometric analysis using the 8-color flow cytometry technique. Peripheral blood samples were incubated with 2 mixtures containing 8 monoclonal antibodies conjugated to different fluorochromes (illustration in Table 1) for 15 min at room temperature. Then, erythrocytes were lysed via a 10-min incubation with an appropriately diluted BD Lysing Solution (Becton Dickinson, San Jose, CA, USA). After lysing the erythrocytes, the samples were rinsed with Cell Wash (Becton Dickinson, San Jose, CA, USA). Stained samples were loaded into a BD FACS Canto II cytometer (Becton Dickinson, San Jose, CA, USA), and 50,000 cells were recorded. Diva 6 software (Becton Dickinson, San Jose, CA, USA) was used for analysis.

### 2.4. Statistical Analysis

The obtained data were checked for completeness and correctness and entered into a database created in MS Excel 2010 and then analyzed using the Statistica 12.0 package. Descriptive statistics were initially calculated (number of groups, percentage fractions for qualitative variables, mean, standard deviation for quantitative variables). Intergroup comparisons were made: For unrelated quantitative variables, after checking the test assumptions (Shapiro–Wilk test, Levene test) using parametric tests (*t*-tests), non-parametric tests (Mann–Whitney U test) were used if the assumptions were not met.For qualitative variables, the following tests were used: Pearson’s Chi2 and Maximum Likelihood.The direction and strength of the association between the two quantitative variables was assessed using the regression equation and Pearson’s linear correlation coefficient or Spearman’s R nonparametric correlation.In all analyses, *p* < 0.05 was considered significant. Each subgroup of patients was analyzed separately, the groups were not intermixed.

## 3. Results

### 3.1. Analysis of Natural Killer Cell Phenotypes in Whole Blood within Groups

#### 3.1.1. Percentage of CD16+ Natural Killer Cells (Antibody Mix 1)

A statistically significantly higher percentage of CD16+ NK cells in the whole blood of patients with advanced hepatic fibrosis/cirrhosis (F3/F4) was demonstrated using antibody mix 1.

#### 3.1.2. Percentage of NKCD16+ Natural Killer Cells (Antibody Mix 2)

A statistically higher percentage of CD16+ natural killer cells was demonstrated in the group of patients with advanced hepatic fibrosis/cirrhosis (F3/F4), assessed in the second sample. The results are presented in Table 2.

#### 3.1.3. Percentage of CD62L+ Natural Killer Cells

A statistically significantly higher percentage of CD62L+ natural killer cells was found in the group of patients with none/mild/moderate fibrosis (F0/F1/F2).

#### 3.1.4. Percentage of CD62L+ CD94++ Cells

A statistically significantly higher percentage of CD62L+ CD94++ NK cells in whole blood was found in the group of patients with none/mild/moderate fibrosis (F0/F1/F2).

#### 3.1.5. Percentage of CD27+ Cells

A significantly higher percentage of CD27+ natural killer cells was found in whole blood from the group of patients with none/mild/moderate fibrosis (F0/F1/F2).

#### 3.1.6. Percentage of CD127+ and CD27+ Natural Killer Cells

A statistically significantly higher percentage of CD27+ CD127+ natural killer cells was found in whole peripheral blood from the group of patients with none/mild/moderate fibrosis (F0/F1/F2).

#### 3.1.7. Percentage of CXCR3+ CD27+ Cells

A statistically significantly higher percentage of CXCR3+ CD27+ NK cells in whole blood was found in the group of patients with none/mild/moderate fibrosis (F0/F1/F2).

Figure 2 presents the results discussed above in a graphical form. Figure 3 presents an exemplary result from the cytometric analysis of peripheral blood with labeled NK cell subpopulations.

### 3.2. Analysis of Intrahepatic NK Cells in Groups

There was no statistically significant difference in the percentage of intrahepatic CD56 cells and their population index between the analyzed groups (results shown in Table 3).

The following figures (Figure 4 and Figure 5A–D) show images of liver biopsies used for analysis, with visible marked AOIs within them. Full-screen close-ups of the AOI and the effect of the work of the counting modules are also shown. Due to the nature of the digitized image, and the possibility of smooth zooming and rotation, the magnifications given in the figures are estimated. 

### 3.3. Analysis of the Correlation between the Percentage of NK Cells in Whole Peripheral Blood and the Liver

#### 3.3.1. Peripheral Blood CD62L+ NK Percentage vs. NK Cell Population Density in the Liver

A positive correlation was found between the percentage of CD62L+ natural killer cells in peripheral whole blood and the population index of intrahepatic CD56+ lymphocytes in both groups (Figure 6).

#### 3.3.2. Percentage of CD94+ NK Cells in Peripheral Blood vs. Percentage of CD56 NK Cells in the Liver

A positive correlation was found between the percentage of NKCD94+ cells in peripheral whole blood and the percentage of intrahepatic CD56+ lymphocytes in both groups (Figure 7)

#### 3.3.3. Percentage of CXCR3+ and CD94+ Natural Killer Cells in Peripheral Whole Blood vs. Population Density of CD56 Lymphocytes in the Liver

A positive correlation was found between the percentage of CXCR3+ CD94+ NK cells in peripheral whole blood and the population density of intrahepatic NK cells in both study groups (Figure 8).

#### 3.3.4. Correlation of CXCR3+ CD94+ NK Percentage in Whole Peripheral Blood a Percentage of Intrahepatic CD56+ Lymphocytes in the Group of Patients with Advanced Liver Fibrosis/Cirrhosis (≥F3)

A positive correlation was found between the percentage of CXCR3+ CD94+ natural killer cells in peripheral blood and the percentage of intrahepatic CD56+ lymphocytes in the group of patients with advanced liver fibrosis (≥F3) (Figure 9).

#### 3.3.5. Correlation of the Percentage of CXCR3+CD94+ in Peripheral Blood and the Population Index of Intrahepatic CD56+ Lymphocytes in the Group of Patients with Advanced Liver Fibrosis (≥F3)

A positive correlation was shown between the percentage of CXCR3+CD94+NK cells in peripheral whole blood and the population index of intrahepatic NK cells in the group of patients with advanced liver fibrosis (≥F3) (Figure 10).

#### 3.3.6. Percentage of CD94+ Natural Killer Cells in Peripheral Whole Blood a Population Index of Intrahepatic Lymphocytes in the Group of Patients with None/Mild/Moderate Fibrosis

A negative correlation was found between the percentage of CD94+ NK cells in peripheral whole blood and the population index of intrahepatic lymphocytes in the patients with none/mild/moderate fibrosis (Figure 11).

#### 3.3.7. Correlation of the Percentage of Cells with the CD94++ NK Phenotype in Peripheral Blood and the Population Index of Intrahepatic Lymphocytes in the Group of Patients with None/Mild/Moderate Fibrosis

A positive correlation was found between the percentage of CD94++ natural killer cells in peripheral whole blood and the population index of intrahepatic lymphocytes in the group of patients with mild or moderate hepatic fibrosis (Figure 12).

## 4. Discussion

NK cells, as an important element of the immune system, have not only a significant impact on controlling HCV infection, but also on the process of liver fibrosis. Until 2017, many research papers were written on various phenomena related to NK cells and their phenotypes, defined on the basis of the surface receptors they present. In 2009, Bonorino et al. showed a positive correlation between NK cells with the NKGD2A+/CD94+ phenotype and the degree of necroinflammatory activity and the extent of fibrosis [31]. 

The fibrosis of the liver parenchyma is strictly dependent on the function of activated stellate cells [32]. The regulation of HSC activity depends on many factors, both humoral and at the level of intercellular interactions. 

In the population of lymphocytes present in the liver, the numerous NK cells present here may have a particularly significant impact on the function of HSCs. In mouse models, NK cells have been shown to selectively eliminate newly activated and senescent stellate cells through the NKp46 receptor [33,34]. The liver has a specific feature in relation to the immune system, which is its tendency to induce immune tolerance. This property is also important in the case of CHC. Muhanna et al. showed an increased number of natural killer cells NKG2A+/CD94+ in the blood of patients with CHC and their reduced cytotoxic function, and Holder et al. demonstrated the ability to inhibit the function of natural killer cells by cells infected with HCV [35,36]. This leads to an increase in the number of NK cells with impaired function in the liver, which is indirectly confirmed by our results. Moreover, the research proved the existence of an additional path of fibrogenesis, which is the activation of stellate cells through the phagocytosis of NK lymphocytes settled in the liver [37]. 

Of course, the immune mechanisms within the liver for fibrosis are not based solely on the destruction of HSCs by NK cells. An important modulatory role is played by CD4+ lymphocytes (Tregs). It was shown, by Weiskirchen et.al, that HCV infection-specific Tregs affect the mechanisms of fibrosis by inhibiting the cytotoxic activity of NK cells against HSCs, coming into direct contact with them. In addition, through IL-10 and IL-15, they inhibit the effector T cells [38]. Recently, other mechanisms have also been described, e.g., those related to IL-33. This interleukin secreted by hepatocytes in a state of “stress” triggers the activation of HSC, also acting profibrogenicly [39]. In addition, it increases the influx of NK cells from peripheral blood to the liver. 

On the other hand, there are reports in the literature about a completely opposite effect of regulatory lymphocytes on fibrosis processes in the liver. Claasen et al. showed that the presence of a large number of highly activated and differentiated influent Tregs re-sult n chronic HCV liver infection. The presence of these lymphocytes may also result in a reduction in the degree of fibrosis [40]. This is due to the inhibition by regulatory cells of the immoderate immune response and secondary damage to the liver parenchyma. Tregs may also negatively regulate the profibrotic roles of Th2 cells and inflammatory monocytes/macrophages that secrete IL-4 and TGF-β, respectively, and, by secreting amphiregulin, may inhibit the development of fibrosis by promoting hepatocyte proliferation. Still, the role of regulatory T cells in the process of liver fibrosis remains to be investigated [41].

The very mechanisms of fibrosis are affected by structural proteins of the virus. Bansal et al. in 2015 described the effect of the non-structural viral protein NS3/4A on liver fibrosis. In the initial phase of infection, this protein, by direct activation of HSC, intensifies fibrosis, and in the chronic phase of inflammation, it reduces the influx of inflammatory cells to the liver, thanks to which the amount of pro-inflammatory chemokines is reduced, and the macrophage phenotype changes to M2 via a suppressive effect. The IL-15 polymorphism has also been shown to be associated with advanced hepatic fibrosis [42]. 

In addition to research on other immune mechanisms, for several years there has been evidence in the literature of the significant effect of NK cells on collagen fibroplasia. In previous studies, Gabriel et al. showed that in patients with rapid progression of fibrosis, the number of NK cells is significantly lower than in patients with low annual progression of fibrosis [43]. In our work, we observed a statistically significantly higher number of NK cells in the peripheral blood of patients with advanced hepatic fibrosis. This feature concerned NK cells with the CD56+ and CD16+ phenotype, and therefore the total number of NK cells (due to the methodology, the cytofluorometric assessment of this cell population was performed twice). Importantly, a much smaller standard deviation of the number of these cells is noticeable in patients with advanced hepatic fibrosis. There was no statistically significant relationship between the number of CD56+CD16+ cells in the peripheral blood and the number and population of NK cells in the liver in both study groups, which may indicate the presence of separate populations of NK cells in the liver and peripheral blood. 

Comparing subsequent NK cell populations between the analyzed groups, a statistically significant difference was found in the percentage of NK cells expressing the CD62L+ receptor. Significantly more of these cells are found in the peripheral blood of patients with a low extent of fibrosis. The study by Peng et al. showed that CD62L+ natural killer cells possess a higher amount of granzyme B and show greater cytolytic activity than CD62L- cells [44]. In addition, this receptor is essential for the accumulation and maturation of NK cells in the liver. In the studies of Bourayou et al., it was also shown that CD62L+ cells show a greater proliferative capacity both in vivo and in vitro; moreover, cells expressing CD62L+ had a greater capacity to produce IFN γ after receptor stimulation [45]. These results seem to be confirmed in our research. The greater the percentage of NK cells with the CD62L+ phenotype in the peripheral blood, the greater the population of NK cells in the liver, regardless of the extent of fibrosis. 

In the group of patients in whom the degree of liver fibrosis was <F3, a higher percentage of NK cells with the CD62L+CD94+ phenotype was observed. Research by Cichocki et al. on NK functions depending on CD94 surface expression showed that the CD62L+CD94+ subtype has a greater ability to penetrate vascular endothelium and produce IFN γ after activation via dendritic cells [46]. Li et al. showed that the number of CD94+ NK cells is higher in patients infected with HCV than in healthy people [47]. These cells have a lower ability to activate dendritic cells; however, due to the association of this mechanism with the HLA-E system, and not HLA-A, it can be presumed that the function of NK cells in relation to stellate cells is not disturbed. 

Another NK cell population with a greater representation in the blood of patients with low-grade fibrosis is the population of CD27+ and CD27+CD127+ cells. Studies by Vossen et al. indicate that CD27 is a receptor present on younger forms of NK cells, with a greater ability to produce cytokines, while CD27 expression disappears after IL-15 stimulation, with CD27-NK cells showing an outstanding cytotoxic function [48]. Similar results were obtained by a research team led by Silva A. [49]. CD27+ NK cells also have a high proliferative capacity in vivo and a much greater sensitivity to chemokines, which is associated with the expression of CXCR3 [50]. The CD27 receptor is also necessary for the proper functioning of NK cells after activation. NK cells of mice lacking CD27 expression after activation secreted less interferon and had less cytotoxic capacity [51]. The CD127 membrane receptor characterizes NK cells with high proliferative activity, and the percentage of these cells is lower in HCV-infected patients compared to healthy individuals [52]. 

The lower percentage of these NK cell phenotypes in patients with advanced liver fibrosis may indicate a lower availability of young forms of NK cells more sensitive to cytokines in the peripheral blood of these patients, and thus less NK cells capable of inhibiting the activity of stellate cells that can migrate from the peripheral blood to the liver. Moreover, analyzing the NK cell phenotypes prevailing in the group of patients with less extensive fibrosis, it seems that this is the intermediate phenotype of CD56int NK cells described by Lima et al., characterized by a wide range of functions [53]. 

In the literature, the role of CXCR3-related chemokines is often implicated with advanced liver fibrosis or cirrhosis. Argirion et al. indicate the correlation of the concentration of CXCR3 ligands—CXCL9, 10 and 11 with the extent of fibrosis—as being several times higher in the liver, with a degree of fibrosis >2 on the Metavir scale. As a result, the number of incoming lymphocytes increases and the damage to the liver parenchyma increases, which translates into a greater extent of fibrosis [54]. In addition, there are studies showing a relationship between the concentration of CXCL10 in the blood serum and the severity of liver fibrosis [55]. Eisenhard et al. described the phenotype of NK cells CXCR3+ CD56bright, which have the highest cytolytic potential after exposure to activated HSC. These cells, in addition to expressing CXCR3, have a significant expression of CD27, CD94, CD62L and NKp44, losing it as they mature, which is associated with a simultaneous loss of cytolytic function toward HSCs. What is important in patients with CHC is that the same cells show impaired function, related with less degranulation and IFNγ secretion. In advanced hepatic fibrosis, the number of these cells in the liver increases, which additionally indicates their lost functions of HSC cytolysis and interferon production. In our study, we found a statistically significantly lower percentage of CXCR3+ CD27+ cells in the peripheral blood of patients, with advanced positive correlation of the population index of CD56 lymphocytes in the liver and the percentage of CXCR3+ CD94+ cells in the peripheral blood in both groups. However, in the group of patients with advanced liver fibrosis or cirrhosis, analyzed separately, this correlation is almost twice as high as in the group of patients with little extent of fibrosis. It concerns both the percentage of CD56+ lymphocytes in the liver and the population index of CD56+ lymphocytes in this organ. This indicates the important role of cytokines in acting on the CXCR3 receptor in activating the migration of NK cells to the liver. This observation is in line with the results of Eisenhard’s analyses [56]. Intrahepatic accumulation of NK cells correlating with the advancement of fibrosis was also described by Nel et al., and these cells were characterized as the NKp44+ CD3-CD56+ phenotype [57]. The above observations indicate that in advanced liver fibrosis there is an intrahepatic accumulation of functionally impaired NK, *inter alia*, in relation to HSC, and thus a decrease in their ability to influence further progression of fibrosis. 

The phenomenon of inverse correlation between fibrosis and NK activity in relation to HSC was also noticed by Kamm et al.; however, they also showed that in vitro NK cells collected from HCV-infected patients show a very strong ability to induce stellate cell apoptosis, which can be inhibited with antibodies against TRAIL, NKG2D and FasL [58]. This observation was also confirmed by Fugier et al., who indicated that the ability to degranulate NK cells depends on the severity of fibrosis [59].

The last observed correlation is the relationship between the population of lymphocytes in the liver and the percentage of NK cells expressing CD94; however, there is a mutual contradiction in the obtained results: with lower expression of CD94, the correlation is negative, and with higher expression, it is positive. It seems that this is a random observation related to the low percentages of these cell phenotypes in peripheral blood. In the existing bibliography, we have not found studies to support this observation.

## 5. Conclusions

After analyzing the presented results, we drew the following conclusions:Patients with CHC with advanced liver fibrosis ≥ F3 have a higher percentage of the total population of CD56+ CD16+ NK cells in peripheral blood compared to patients with mild or moderate fibrosis.There is a higher percentage of NK cells with CD62L+, CD94+, CD27+, CD127+ and CXCR3+ phenotypes in the peripheral blood of patients with mild or moderate fibrosis in relation to patients with advanced liver fibrosis or cirrhosis. This indicates a lower availability of functionally active NK cells in the peripheral blood of patients with advanced fibrosis.In patients with CHC, there is a positive correlation between the percentage of NK cells with CD62L+ and CD62L+ CD94+ phenotypes in peripheral blood and the population index of intrahepatic NK cells, regardless of the extent of fibrosis.The percentage of NK cells with the CXCR3+ CD94+ phenotype in the peripheral blood was correlated with the population index and the percentage of NK cells in the liver, with the correlation increasing depending on the degree of fibrosis.In patients with CHC with advanced liver fibrosis, there is intrahepatic accumulation of functionally impaired natural killer cells, thus limiting their influence on further progression of fibrosis.

However, it should be remembered that the study was conducted on a small group of patients. We were able to recruit only 15 patients with a level of liver fibrosis ≥ F3, which does not allow us to draw broad conclusions. In addition, the phenotype of the NK cell subpopulations we analyzed may change over the course of the disease and under the influence of factors other than liver damage. The involvement of the immune system in the process of liver fibrosis and the interrelationships between immune cells, hepatocytes and stellate cells as well as a number of molecular pathways remain to be elucidated. Further research may improve the prognosis and quality of life of patients with hepatitis C.

## Figures and Tables

**Figure 1 diagnostics-13-02187-f001:**
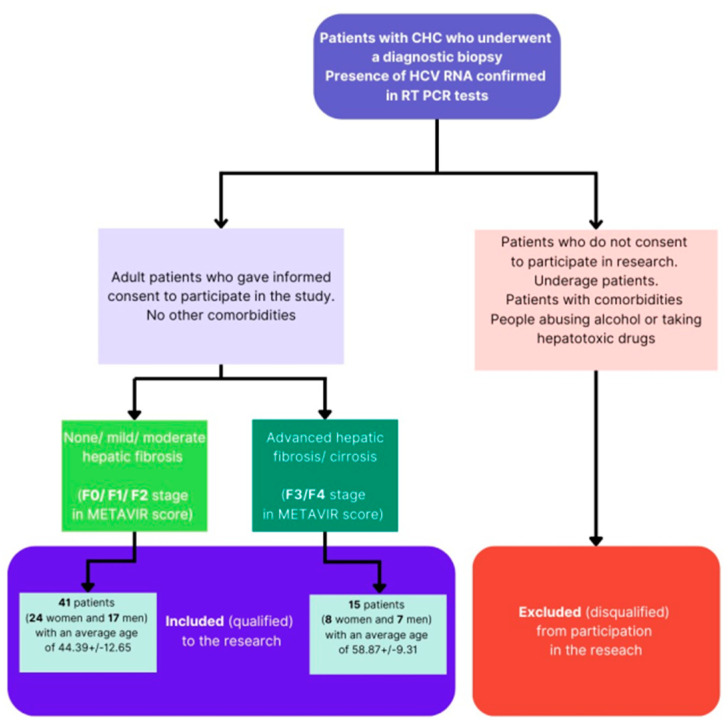
Flowchart explaining patient inclusion/exclusion criteria.

**Figure 2 diagnostics-13-02187-f002:**
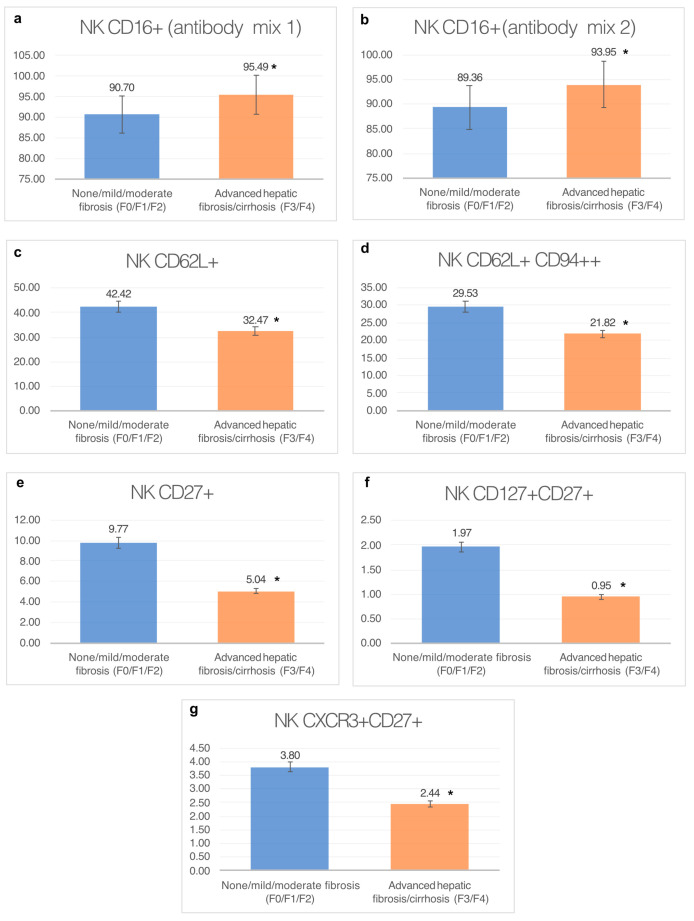
Results of the study of the percentage of different NK cell subpopulations in the groups of patients with none/mild/moderate liver fibrosis (F0/F1/F2) and with advanced hepatic fibrosis or cirrhosis (F3/F4). The graphs show: (**a**) percentage of CD16+ natural killer cells in whole blood from groups (antibody mix 1); (**b**) percentage of CD16+ natural killer cells in whole blood from groups (antibody mix 2); (**c**) mean percentage of CD62L+ natural killer cells in whole blood in the analyzed groups; (**d**) mean percentage of CD62L+ CD94++ natural killer cells in whole blood in the analyzed groups; (**e**) mean percentage of CD27+ natural killer cells in whole blood in the analyzed groups. (**f**) mean percentage of CD27+ CD127+ natural killer cells in whole blood in the analyzed groups; (**g**) mean percentage of CXCR3+ CD27+ natural killer cells in whole blood in the analyzed groups. * Statistically significant difference in the percentage of the NK cell population between analyzed groups; *p* < 0.05.

**Figure 3 diagnostics-13-02187-f003:**
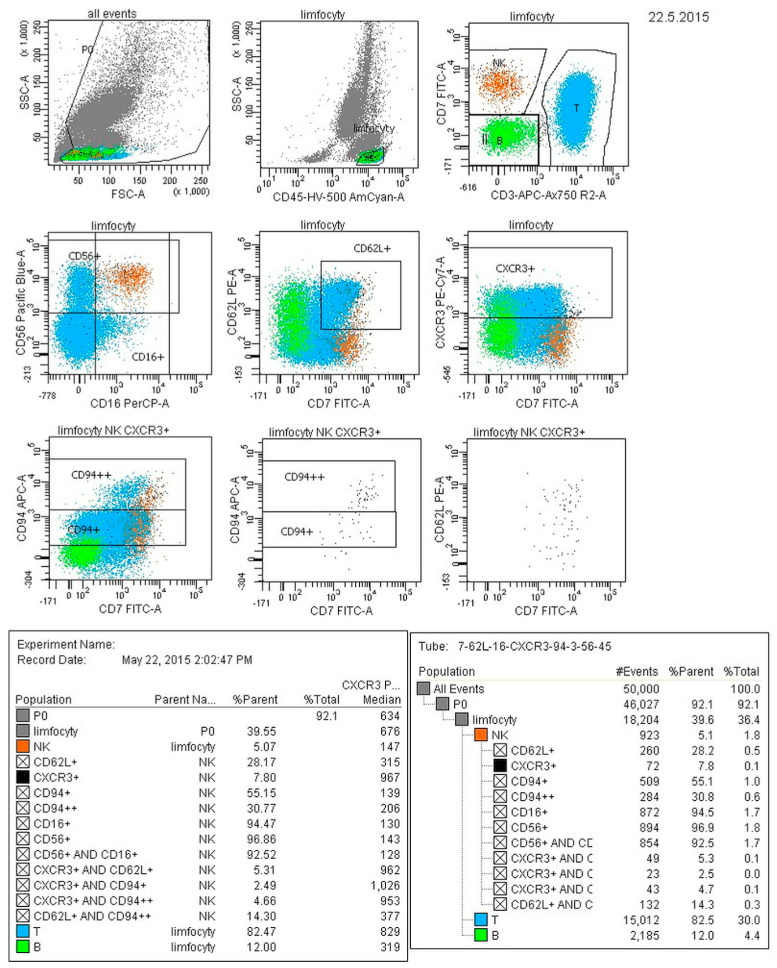
Sample results of the cytometric analysis with specified NK cell phenotypes.

**Figure 4 diagnostics-13-02187-f004:**
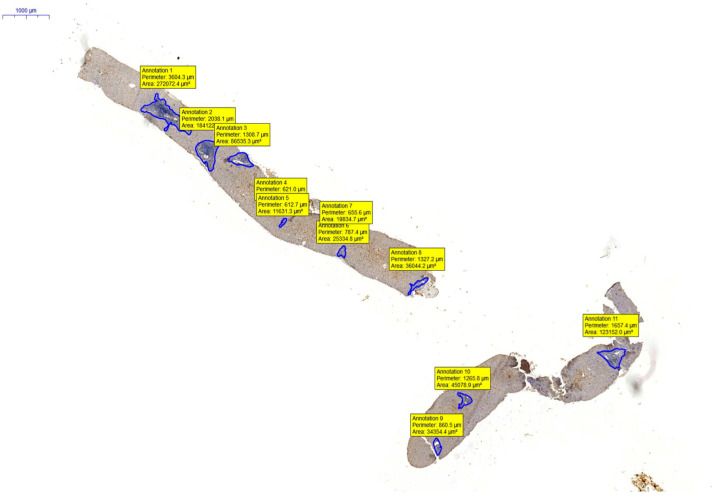
Liver biopsy with marked AOI. In the frames, you can see the measurement values of the area of the selection. Magnification approx. 120×.

**Figure 5 diagnostics-13-02187-f005:**
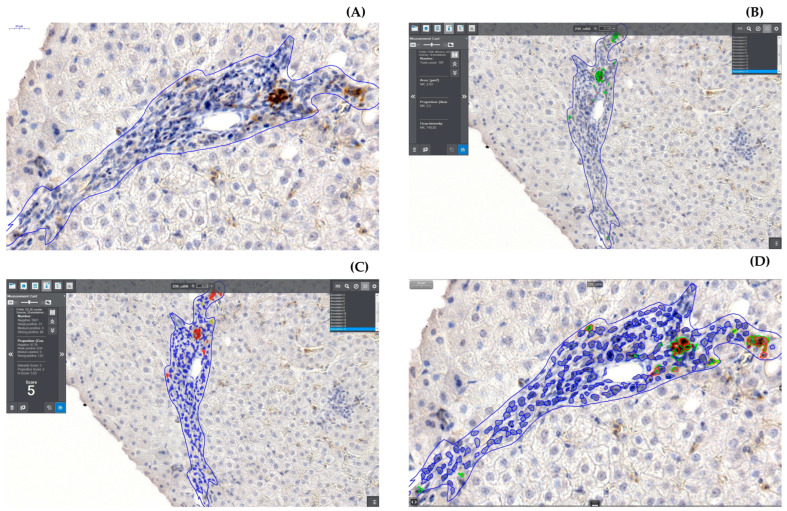
(**A**) Portal space with marked AOI (blue line). Magnification approx. 500×. (**B**) AOI from the previous figure with the output of the HistoQuant module. CD56 cells marked in green. Magnification approx. 430×. (**C**) The same AOI with the output of the NuclearQuant module. Blue indicates portal space lymphocytes, yellow and red CD56 lymphocytes. Magnification approx. 430×. (**D**) AOI from the previous figure with the result of both counting modules superimposed. Blue contour—lymphocytes; red, yellow and green contours—CD56 cells. Visible complementation of the work of the counting modules. Magnification approx. 480×.

**Figure 6 diagnostics-13-02187-f006:**
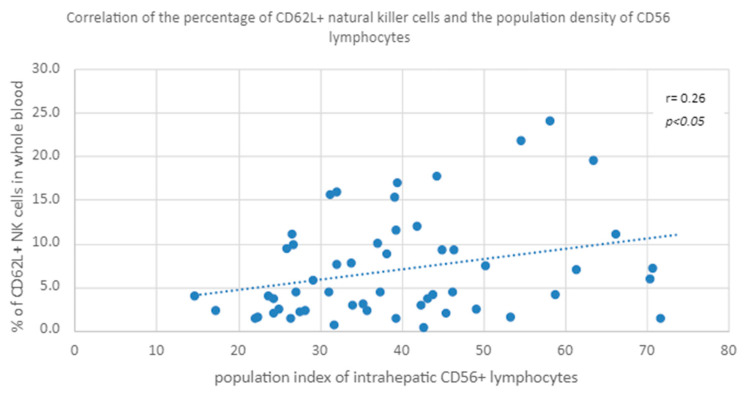
Correlation of the percentage of CD62L+ natural killer cells and the population density of CD56 lymphocytes.

**Figure 7 diagnostics-13-02187-f007:**
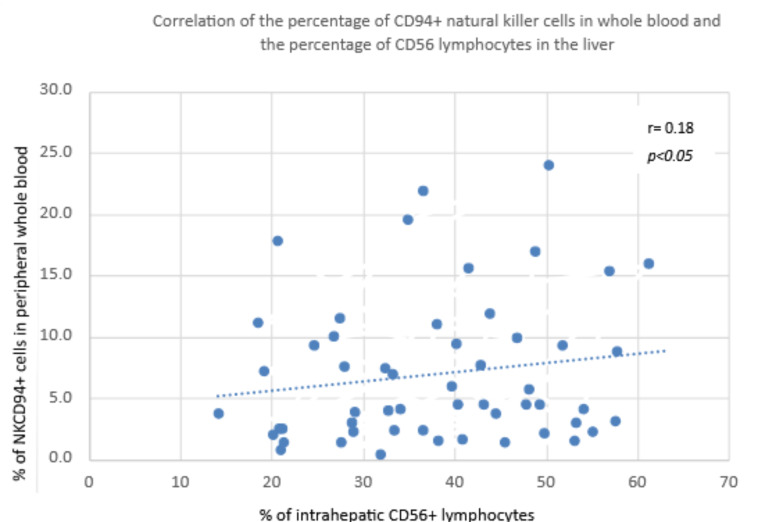
Correlation of the percentage of CD94+ natural killer cells in whole blood and the percentage of CD56 lymphocytes in the liver.

**Figure 8 diagnostics-13-02187-f008:**
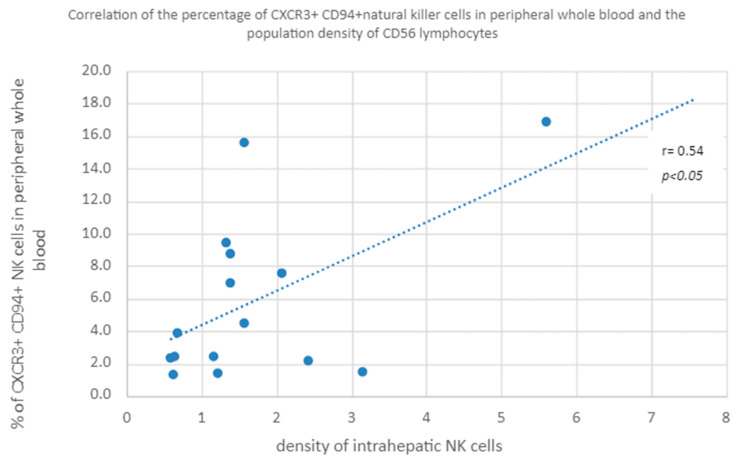
Correlation of the percentage of CXCR3+ CD94+ natural killer cells in peripheral whole blood and the population density of CD56 lymphocytes.

**Figure 9 diagnostics-13-02187-f009:**
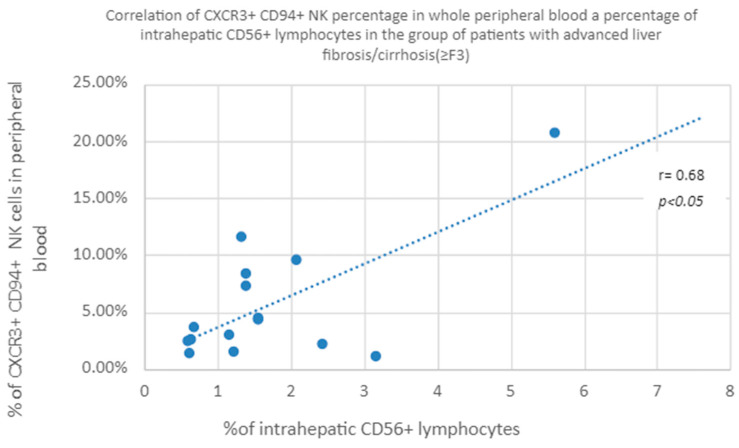
Correlation of the number of CXCR3+ CD94+ natural killer cells in peripheral whole blood and the percentage of intrahepatic CD56+ lymphocytes in the group of patients with advanced liver fibrosis (≥F3).

**Figure 10 diagnostics-13-02187-f010:**
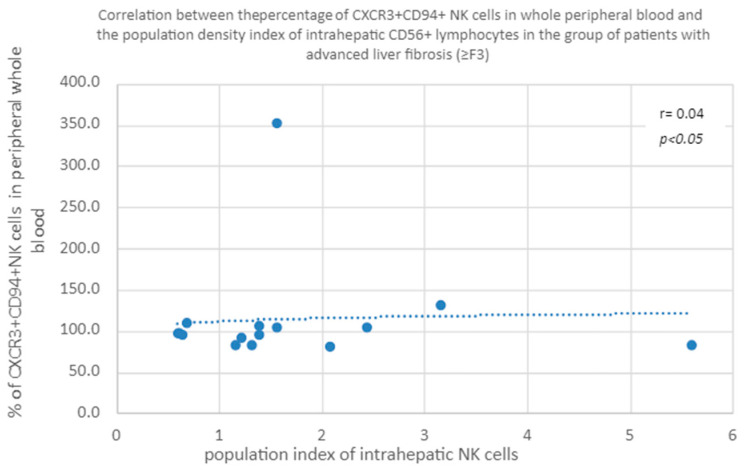
Correlation between the percentage of CXCR3+ CD94+ NK cells in whole peripheral blood and the population density index of intrahepatic CD56+ lymphocytes in the group of patients with advanced liver fibrosis (≥F3).

**Figure 11 diagnostics-13-02187-f011:**
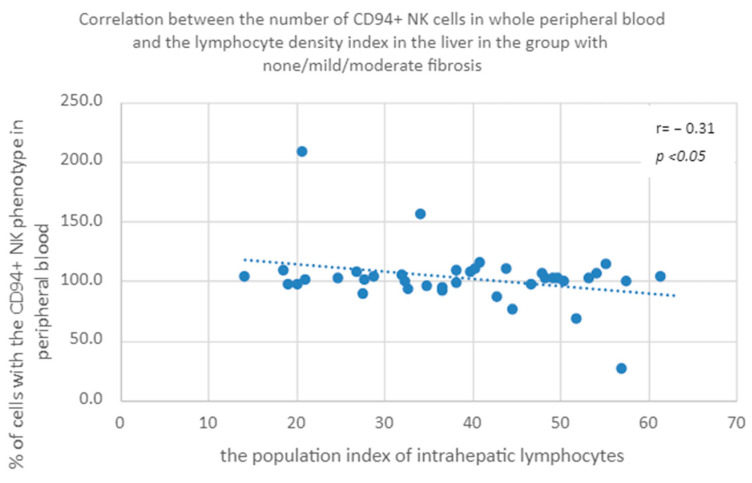
Correlation between the percentage of CD94+ NK cells in peripheral whole blood and the population index of intrahepatic lymphocytes in the patients with none/mild/moderate fibrosis.

**Figure 12 diagnostics-13-02187-f012:**
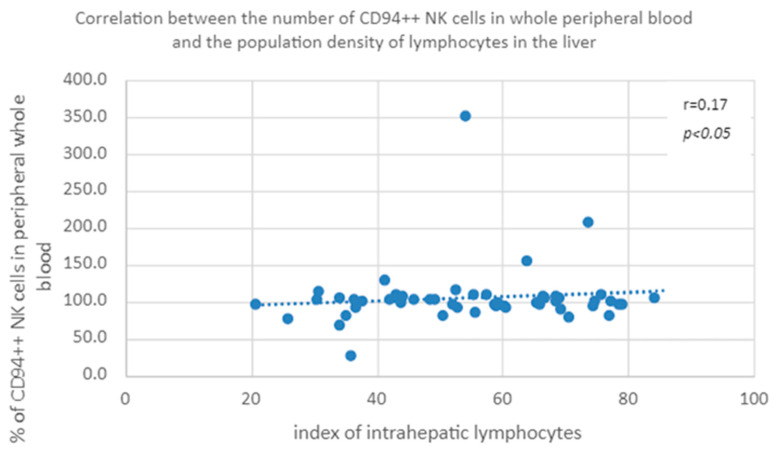
Correlation between the number of CD94++ NK cells in whole peripheral blood and the population density of lymphocytes in the liver.

**Table 1 diagnostics-13-02187-t001:** Panel of antibodies used for assays.

	FITC	PE	PerCP-Cy5.5	PE-Cy7	APC	APC-Ax750	Pacific Blue	V500
1	CD7	CD62L	CD16	CXCR3	CD94	CD3	CD56	CD45
2	CD7	CD127	CD16	CXCR3	CD27	CD3	CD56	CD45

Abbreviations of fluorochrome names: FITC—fluorescein isothiocyanate; PE—phycoerythrin; PerCP-Cy5.5—peridininochlorophyll-cyanin 5.5; PE-Cy7—phycoerythrin-cyanin 7; APC—allophycocyanin; APC-Ax750—allophycocyanin Ax750 for CD3 visualization; Pacific Blue—1,2-ditetradecanoyl-sn-glycero-3-phosphoethanolamine, triethylammonium salt; V500 (Horizon V500)—violet-excitable dye.

**Table 2 diagnostics-13-02187-t002:** Percentage of CD16+ natural killer cells in whole blood from groups using antibody mix 1 and antibody mix 2. * Statistically significant difference in the percentage of the NK cell population between the analyzed groups; *p* < 0.05.

NK CD16+		*n*	Mean Value ± Standard Deviation
None/mild/moderate fibrosis (F0/F1/F2)	antibody mix 1	41	90.70 ± 6.69 *
	antibody mix 2		89.36 ± 6.84 *
Advanced hepatic fibrosis/cirrhosis (F3/F4)	antibody mix 1	15	95.49 ± 1.87 *
	antibody mix 2		93.95 ± 3.56 *
Total in the group	antibody mix 1	56	91.98 ± 6.17
	antibody mix 2		90.64 ± 6.42

**Table 3 diagnostics-13-02187-t003:** Mean percentage of intrahepatic CD56 and mean population index (CD56 number per 10,000 μm^2^) cells in the analyzed groups.

Group of Patients	*n*	Analyzed Parameter	Mean Value ± Standard Deviation
None/mild/moderate fibrosis (F0/F1/F2)	41	% of intrahepatic CD56 cells	8.12 ± 0.10
		population index of intrahepatic CD56	7.02 ± 6.12
Advanced hepatic fibrosis/cirrhosis (F3/F4)	15	% of intrahepatic CD56 cells	5.61 ± 0.05
		population index of intrahepatic CD56	5.82 ± 5.05
Total in the group	56	% of intrahepatic CD56 cells	7.45 ± 0.09
		population index of intrahepatic CD56	6.69 ± 5.73

## Data Availability

The data presented in this study are available on request from the corresponding author.

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
