# Peer review of "Association of NK Cells with the Severity of Fibrosis in Patients with Chronic Hepatitis C"

_diagnostics, 2023, doi:10.3390/diagnostics13132187_

Round 1
Reviewer 1 Report
The authors assessed the association of NK cell phenotypes and the severity of hepatic fibrosis, as determined by liver biopsy, in 56 HCV-infected patients. They found that certain NK cell subtypes were associated with the advanced hepatic fibrosis or cirrhosis (≥ F3) in patients with HCV.
1. The abstract and the manuscript: please avoid the terms of low or advanced hepatic fibrosis. Trim them all, and replace with “divided into patients with/without advanced hepatic fibrosis or cirrhosis (≥ F3) to avoid reader confusion.
2. Line 98: what did “CHP” mean?
3. Line 111: please merge the section “study group” and “control group” into one. I strongly discourage the use of terms “study and control group”. Just simply state the inclusion and exclusion criteria. State clearly the definition of none, mild, moderate, advanced hepatic fibrosis and cirrhosis as F0, F1, F2, F3, and F4 by Scheuer and METAVIR scores. Define the aim of the study to set a cut-off stage of ≥ F3 to be advanced hepatic fibrosis or cirrhosis. Lines 119-125 seem to be the exclusion criteria. Please rewrite subsection 2.1.1 and 2.1.2 as one subsection and redefine all the related term accordingly.
4. Number the Tables with Arabic instead of Romanic symbols (Table 1, 2, 3; not Table I, II, III).
5. The authors should state the limitations of the study, (1) limited case numbers in this study, particularly those with ≥ F3; (2) cross-section study, thus the causal relationship of NK subtypes and the severity of hepatic fibrosis may be hard to be established.
Nil
Author Response
On behalf of myself and the co-authors, I would like to thank you for all your valuable comments regarding the work "Association of NK cells with the severity of fibrosis in patients with chronic hepatitis C". Your involvement in the review of the work allowed us to improve the quality of our experiment.
According to your suggestions:
- We have changed the descriptions of the severity of fibrosis. The term "low/advanced liver fibrosis" could be confusing, so we replaced it with the definitions suggested by the reviewer. Moreover, in the introduction, we tried to describe the liver fibrosis scales used.
- Line: 98: the abbreviation CHP is a typo. The term CHC (Chronic Hepatitis C) should be included in the text. Thank you for paying attention.
- Subsections 2.1.1 and 2.1.2 have been rewritten and the groups of Patients involved in the experiment described in detail. In addition, we have added a graphic (Figure 1. Flowchart explaining patient inclusion/exclusion criteria), which we hope will clearly present the criteria for inclusion and exclusion of Patients from the study.
- All Tables are described using Arabic symbols.
- In the conclusions we tried to emphasize the preliminary nature of our research. Of course, a small group of patients does not allow for the formulation of certain conclusions, and the process of liver fibrosis is so complicated that simple links between NK subtypes and the severity of liver changes are insufficient. Our experience will continue as directed by the Reviewers.
Once again, I express my sincere gratitude for taking the trouble to review.
Regards,
Reviewer 2 Report
The topic of the study is important, the results on the study of the role of NK in the development of liver fibrosis in patients with chronic hepatitis C are numerous, but often contradictory. The authors received a number of interesting data, but the presentation of data is unacceptable. Most correlations are very weak. The design of the study also raises questions. The manuscript is very difficult to read due to the incorrect presentation of data.
Main remarks
1. It is necessary to indicate when the work was performed, since there are currently other WHO clinical recommendations. Thus, the diagnosis of liver fibrosis is recommended to be carried out by non-invasive methods, and treatment with the use of DAA.
2. The description of patients requires changes. For example, what was the difference between the exclusion parameters for the study group and the reference group patients? It is necessary to present a separate table with the characteristics of patients in order to understand how many patients with a certain stage of fibrosis, the level of liver enzymes, etc. What is the difference between reference group (line 102) and control group (line 126)?
3. Edit figures 1-3 according to the rules of the journal, bring in the Results section.
4. Section 2.2.3. Digitalization and digital analysis is described in too much detail.
5. Section 2.3. Cytofluorometric analysis: what was done first, stained peripheral blood with antibodies or lysed erythrocytes? (lines 223-228).
6. Table 1 is framed incorrectly. Not all abbreviations are deciphered.
7. The data presentation is unacceptable. Why is there the same Сount for different populations everywhere in Tables II-VIII? Combine the data of these tables, as well as sections 3.1.1. - 3.1.7.
8. Section 3.2. Analysis of intrahepatic NK cells in groups: quantitative data are not provided.
9. Tables IX-XV – correlations are usually expressed graphically.
10. All NK subpopulations used in the study and their biological significance should be described in the Introduction or Materials and Methods.
11. Different designations of the same NK subpopulations were used in the Results and in the Discussion.
12. Lines 372-373. No data is provided that Tregs’ role in the development of hepatic fibrosis appears controversial. High levels of intrahepatic Treg cells are correlated with less extensive fibrosis [PMID: 20129690].
13. Introduction: lines 81-90 – no references, insert.
14. Typos and incorrect terms: line 98- CHP ?
What does the abbreviation AOI mean?
Line 18 - infected with type C hepatitis virus;
Line 119 - normal fibrosis?
Author Response
On behalf of myself and the co-authors, I would like to thank you for all your valuable comments regarding the work "Association of NK cells with the severity of fibrosis in patients with chronic hepatitis C". Your involvement in the review of the work allowed us to improve the quality of our experiment. In accordance with the comments, we tried to change the way of presenting the results of the experiment - so that the article was more readable and understandable. We also prepared particular characteristics of the Patients and tried to explain in detail how and why the experiment was designed.
- The experiment lasted several years and began in 2013, with the preparation of the experiment plan and obtaining the consent of the local Bioethics Committee. The patients recruited for the experiment were undergoing the diagnosis of liver fibrosis, which included a number of laboratory and imaging tests, but in order to confirm the diagnosis and assess the severity of the disease, they were qualified for a biopsy. The patients we describe were treated with interferon and ribavirin, but contemporary therapeutic strategies are different - we have described them in the Material and Methods chapter.
- We have prepared a new description of the characteristics of patients qualified for the study. We tried to clearly explain the inclusion and exclusion criteria, as well as the classification of patients according to the stage of fibrosis. We also present a proposal for a graphical representation of the study groups.
- Figures 3-5 were compiled as required by the journal and transferred to the Results section.
- As suggested, we tried to shorten chapter 2.2.3 Digitalization and digital analysis.
- In the description of the methodology of cytometric assay preparation we emphasized that erythrocytes were lysed after labeling the subpopulation of cells – then the probes were washed using EasyLyse™ Erythrocyte-Lysing Reagent.
- We improved the quality of Tables and Figures and adapted them to the editorial requirements of the journal. We have expanded all the abbreviations listed in Table 1.
7/9.We propose a different way of presenting the results. The data from the table are presented in the graphs (Figure 6-12).
We wanted to acknowledge that during the reprocessing of the results, we detected an important typo in the description of the results. In section 3.3.3-3.3.5 we misdescribed the phenotype. Correlation analysis was performed for the subpopulation of CXCR3+CD94+ not of CXCR3+CD94++. Thanks to the Reviewer's comments, we were able to catch this important error, for which we are sincerely grateful
- We have added quantitative data to Section 3.2.
10/11. A brief description of the NK cell subpopulation with their characteristic phenotype was described in the introduction. The differentiation of NK cells and their comprehensive participation in many immunological processes could be the subject of a separate paper (we are considering preparing a review paper). We also tried to unify the description of the studied NK cell subpopulations but if the Reviewer has specific comments as to the description of the phenotype of the subpopulation of the examined cells, we kindly ask you to highlight the paragraph/line of text to which we must pay special attention.
- The role of regulatory T lymphocytes in the process of liver fibrosis is of course extremely complex - in accordance with the Reviewer's valuable comment, we tried to emphasize that literature reports also indicate the inhibition of fibrogenesis by Tregs.
- We have added references 21-25 to support the data presented in this section.
- Thank you for catching all typos and mental shortcuts. They have been corrected.
Once again, I express my sincere gratitude for taking the trouble to review.
Regards,
Reviewer 3 Report
Dear Editor,
Kleczka et al., in the study evaluated the relationship between the number and phenotype of NK cell subsets in peripheral blood (PB) and total NK cell percentage, population density and the degree of liver fibrosis of patients infected with type C hepatitis virus (HCV+). I have few comments:
- Introduction section: please add references in this section between line 81 – 90.
- Materials and Methods section: please, insert a flowchart to clarify the enrollment and the inclusion/exclusion criteria of patients, study group and control group. Additionally, please clarify if the three groups are intermixed and the decision to create a reference group.
- 2.3. Cytofluorometric analysis sub-section: Figure 3 was not included in the main text of this sub-section. I suggest to insert the result of cytometric analysis with specified NK cell phenotypes in the correct section of paper (Results section).
- The conclusions need a better explanation. The several points if possible should be related to each other.
Kind regards
Minor editing of English language required
Author Response
Regards,
On behalf of myself and the co-authors, I would like to thank you for all your valuable comments regarding the work "Association of NK cells with the severity of fibrosis in patients with chronic hepatitis C". Your involvement in the review of the work allowed us to improve the quality of our experiment.
According to your suggestions:
- We have extended the description of the theoretical assumptions of the experiment. We have presented a brief characterization of scales evaluating liver fibrosis and the role of various natural killer cell subpopulations in organ remodeling. We have added references to paragraphs 81-90 to confirm the data contained therein.
- We have edited subsections 2.1.1. and 2.1.2. We prepared a new description of the study groups and, as suggested by Reviewer, presented the Patient characteristics in a graphic (Figure 1. Flowchart explaining patient inclusion/exclusion criteria)
- The figure showing an example result of the cytometric analysis has been moved to the "Results" chapter. Other results are available in the database (link provided in the "Supplementary materials" chapter) or from the corresponding author.
- In the "Conclusions" chapter, we tried to present the limitations of our experiment, and also to relate the obtained results.
Once again, I express my sincere gratitude for taking the trouble to review.
Round 2
Reviewer 2 Report
The authors have significantly improved the manuscript, but the presentation of the results requires further correction.
Line 295 - CD3 visualisation.
All Tables require stylistic editing. Denote the number of patients not by “count”, but by the generally accepted notation “n=41” and “n=15”. The results should be presented as mean ± SD, and statistically significant differences should be indicated. Tables 2 and 3 should be combined. The names of Tables 2 and 3 are the same. Table 3 uses mix 2 - correct. Also, in Table 3, the wrong figure is to correct 9395 to 93.95.
Tables 2-8 can be excluded, and instead present Figure 2, the results of which are more visual.
Figures 6-12: denote the axes, write the correlation coefficients r or R and the statistical significance of the correlations.
Author Response
Reply to the Review Report Round 2
On behalf of myself and the co-authors, I would like to thank you once again for all the valuable comments that have been provided to us. According to the recommendations:
- The type of antigen labeled with APC-Ax750 has been corrected.
- Tables 2 and 3 have been edited, their descriptions have been corrected, and the results of the labeling of antibody mix 1 and 2 combined are presented.
- We resigned from Tables 4-8, the results presented in them are graphically illustrated in Figure 2.
- Graphs showing correlations between individual cell subtypes have been described taking into account r and p values.
Thanks to the comments of Reviewer 2, we were able to improve the quality of the presentation of our results and we hope that the work "Association of NK cells with the severity of fibrosis in patients with chronic hepatitis C" will be submitted for publication.
Thank you very much for any help
Regards,
Anna Kleczka and co-authors